# Interstitial Lung Disease Associated with Lung Cancer: A Case–Control Study

**DOI:** 10.3390/jcm9030700

**Published:** 2020-03-05

**Authors:** Quentin Gibiot, Isabelle Monnet, Pierre Levy, Anne-Laure Brun, Martine Antoine, Christos Chouaïd, Jacques Cadranel, Jean-Marc Naccache

**Affiliations:** 1Service de Pneumologie, Centre Hospitalier Intercommunal de Créteil, 94000 Créteil, France; quentin.gibiot@chicreteil.fr (Q.G.); christos.chouaid@chicreteil.fr (C.C.); 2Faculté de Médecine, Université Paris-Est Créteil, 94000 Créteil, France; 3Institut Pierre-Louis de Santé Publique (EPAR Team) et Sorbonne Université, Department de Santé Publique, APHP, Hôpital Tenon, 75020 Paris, France; pierre.levy@aphp.fr; 4Service de Radiologie, APHP, Hôpital Cochin, Paris 75014, France, Faculté de Médecine, Université Paris Descartes, 75020 Paris, France; 5Service d’Anatomie et Cytologie Pathologique, APHP, Hôpital Tenon, 75020 Paris, France; martine.antoine@aphp.fr; 6Service de Pneumologie, Site constitutif du centre de référence des maladies pulmonaires rares OrphaLung, APHP, Hôpital Tenon, 75020 Paris, France; jacques.cadranel@aphp.fr; 7Faculté de Médecine, Université Paris Descartes, 75020 Paris, France

**Keywords:** interstitial lung disease, lung cancer, prognosis, retrospective, outcomes

## Abstract

Interstitial lung disease (ILD) seems to be associated with an increased risk of lung cancer (LC) and to have a poorer prognosis than LC without ILD. The frequency of ILD in an LC cohort and its prognosis implication need to be better elucidated. This retrospective, observational, cohort study evaluated the frequency of ILD among LC patients (LC–ILD) diagnosed over a 2-year period. LC–ILD patients’ characteristics were compared to those with LC without ILD (LC–noILD). Lastly, we conducted a case–control study within this cohort, matching three LC–noILDs to each LC–ILD patient, to evaluate the ILD impact on LC patients’ prognoses. Among 906 LC patients, 49 (5.4%) also had ILD. Comparing LC–ILD to LC–noILD patients, respectively, more were men (85.7% vs. 66.2%; *p* = 0.02); adenocarcinomas were less frequent (47.1% vs. 58.7%, *p* = 0.08); median [range] and overall survival was shorter: (9 [range: 0.1–39.4] vs. 17.5 [range: 0.8–50.4] months; *p* = 0.01). Multivariate analysis (hazard ratio [95% confidence interval]) retained two factors independently associated with LC risk of death: ILD (1.79 [1.22–2.62]; *p* = 0.003) and standard-of-care management (0.49 [0.33–0.72]; *p* < 0.001). Approximately 5% of patients with a new LC diagnosis had associated ILD. ILD was a major prognosis factor for LC and should be taken into consideration for LC management. Further studies are needed to determine the best therapeutic strategy for the LC–ILD population.

## 1. Introduction

Lung cancer associated with interstitial lung disease (LC–ILD), particularly idiopathic pulmonary fibrosis (IPF), has been described for many years [1]. IPF is the most serious ILD, occurring predominantly in male smokers ≥60 years old, with a median survival of 2–4 years [2]. Other ILDs essentially involve inflammatory processes and carry better prognoses. LC–ILDs could result from several risk factors common to LC and ILD, i.e., smoking and asbestos exposure [3,4]. Pertinently, IPF pathophysiology resembles that of cancer development, mainly including epithelial cell anomalies ranging from metaplasia to carcinomatous transformation [5]. Moreover, the chronic inflammation found in other ILDs is thought to promote genetic mutations leading to carcinogenesis [6].

ILD frequencies reported in LC populations ranged between 2.4% and 10.9% [7], while IPF patients’ LC risk was evaluated at 4.96– and 7.3–times higher in two case–control studies [7,8,9]. Furthermore, LC-associated mortality was more than seven times higher for patients with IPF [10].

LC–ILD patients are mostly men, older and heavy smokers [11,12,13]. Their LCs are more frequently located in the lower lobes and fibrotic areas, and are more often squamous-cell carcinomas [7]. The results of some studies suggested that LC–ILD tumors less frequently carried oncogenic driver mutations, probably because most patients were active or former smokers [14]. LC–ILD patients’ prognoses, evaluated in a few studies, seems to be worse compared to those with LC without ILD (LC–noILD) [15,16].

Most of the above-cited LC–ILD studies were conducted on IPF patients from Asia (Japan and Korea), and were single-center, surgical series. Indeed, little information is available on Caucasian patients with LC–ILD, subclinical ILD or advanced LC [12,17,18].

This study was undertaken to evaluate the frequency and clinical characteristics of LC–ILD in a cohort of non-selected Caucasian LC patients and the impact of ILD presence on LC prognosis.

## 2. Methods

This retrospective, observational study was conducted at two French university hospitals. The study was approved by the Institutional Review Board of the French Society of Respiratory Medicine (CEPRO 2016-007); informed consent was not required for this observational study according to French law.

### 2.1. Patient Selection

All consecutive patients discussed in multidisciplinary meetings between January 2012 and December 2013 were screened. Patients diagnosed with LC were included when they had histologically confirmed LC and a baseline chest high-resolution computed-tomography (CT) scan.

### 2.2. Data Collection

The following information was collected from each patient’s medical records: age, sex, smoking status, professional asbestos exposure, histology (according to the World Health Organization (WHO) histological classification), stage (based on the seventh-edition of the international TNM cancer-staging criteria), presence of oncogenic driver mutation, presence of ILD, pulmonary function test’s (PFTs) for patients with ILD, LC therapeutic management, overall survival (OS) and progression-free survival (PFS) [19,20].

ILD was diagnosed according to ATS/ERS criteria [21]. Two experts (JMN, ALB), blinded to patients’ characteristics, reviewed all baseline CT scans for quality, ILD pattern, presence of emphysema and pleural thickness assessment. The Silva scoring system was applied to evaluate ILD radiological severity (see Appendix A) [22]. The definitive ILD diagnosis was established during a multidisciplinary discussion.

### 2.3. Study Design

We first conducted an epidemiological study comparing LC-ILD and LC-noILD patients. Nine hundred and six patients were included, respectively 49 and 857. Controls were then matched to case subjects according to sex, cancer staging and histology. They were selected from among those included in the general comparative study described above. For each case, three controls were selected, excepted for two cases, for whom only two perfectly matched controls could be found. The study design is summarized in Figure 1.

### 2.4. Statistical Analyses

LC–ILD and LC–noILD patients’ quantitative variables, expressed as mean ± standard deviation (SD) or median (range), as appropriate, were compared with Student’s *t*-test or the Mann–Whitney U-test; their qualitative parameters, expressed as *n* (%), were compared with chi-square tests or Fisher’s exact. Based on those results, a case–control study (three controls matched to each LC–ILD case) was undertaken to evaluate ILD’s impact on LC therapeutics and prognoses.

LC–ILD and LC–noILD survival rates were estimated with the Kaplan–Meier method and compared with log-rank test. Prognostic factors were subjected to univariate and multivariate analyses using a descending stepwise Cox model. Candidate variables were all non-redundant variables with *p* ≤ 0.2. All statistical analyses were computed with Statview software. A two-sided *p*-value of 0.05 defined significance.

## 3. Results

### 3.1. ILD Frequency and Characteristics

Among the 906 consecutive LC patients included, 49 (5.4%) had associated ILD. Their mean age was 66.4 ± 8.8 years and 85.7% were male. Most LC–ILD patients were smokers (91.8%), with mean exposure of 44.4 ± 22 pack-years; 40% were ex-smokers and 60% were current smokers. LC development involved fibrotic areas in 29 (59.2%) LC–ILD patients.

Only 39% of ILDs were noted in the patients’ files by a pulmonologist or radiologist. Autoantibody serology results were available for 14 (29%) patients, BAL fluid analysis for four (8%) and pulmonary function tests for 38 (78%).

Radiological analysis found CT-scan slice thickness of ≤1.5 mm for all but four patients. Radiological patterns were definite usual interstitial pneumonia (UIP) (*n* = 10, 20.4%), probable UIP (*n* = 16, 32.7%), indeterminate for UIP (*n* = 11, 22.4%) and inconsistent with UIP (*n* = 12, 24.5%). Forty-two (85.7%) patients had emphysema and three (6.1%) had pleural plaques. Silva scoring system grade was mild at 1.61 ± 0.5, corresponding to an interstitial features’ extent of ~5%.

The ILD diagnoses retained, based on multidisciplinary discussions, were: 19 IPF (nine definite diagnoses, 10 likely diagnosis-probable radiological UIP without any cause or autoimmunity), 20 unclassifiable due to missing data (principally exposure and immunological screening), four associated connective tissue diseases, three pneumoconioses, and one for each hypersensitivity pneumonia, NSIP or sarcoidosis. ILD diagnosis was based on pathological analysis in 11 of the 13 patients who undergone surgery for LC. The diagnoses in these patients were UIP (*n* = 2), NSIP (*n* = 2), pneumoconiosis (*n* = 3), hypersensitivity pneumonia (*n* = 1) and unclassifiable (*n* = 3). The PFT results, available for 38 (79%) patients, confirmed mild impairment with vital capacity (VC), total lung capacity (TLC), forced expiratory volume in 1 s (FEV_1_) and FEV_1_/VC of 90.9% ± 18.3%, 91.6% ± 17.2%, 78.9% ± 21.7%, 68.8% ± 11.6%, respectively. Ten patients had a restrictive pattern (TLC < 80%), 18 patients an obstructive pattern (FEV_1_/VC < 70%) and four a mixed pattern. Lung diffusing capacity for carbon monoxide, determined for 32 (65.3%) patients, was low (<70%) in 90.6% of them, with the mean ± SD at 55.8% ± 17.5%.

Only three patients received ILD’s treatment, mainly immunosuppressive treatments. None received antifibrotic treatment and none were on supplemental oxygen at the time of LC diagnosis.

### 3.2. LC–ILD and LC–noILD Patients’ Comparisons

LC–ILD patients, compared to LC–noILD, were significantly more frequently men (85.7% vs. 66.2%), with a non-significant trend for less frequent adenocarcinomas (47.1% vs. 58.7%), while ages, smoking histories, asbestos exposures and LC stages were comparable (Table 1). Lung cancer location in LC-ILD patients was the lower lobes in 23 patients, the upper lobes in 23 patients, the middle lobe in one patient and multifocal in two patients. Lung cancer was located in the fibrotic area in 29 (59.2%) LC-ILD patients. Lung cancer location was not systematically reported for LC-noILD so that we couldn’t compare locations between the two groups.

### 3.3. Case-Control Study

The characteristics of the 49 LC–ILD cases were compared to those of 145 matched controls (Table 2). Epidemiological features (sex, age, tobacco use, body mass index, WHO performance status (PS), comorbidities) and cancer parameters (histology, stage) were similar (all *p* > 0.05).

The treatments received for each LC stage are summarized in Appendix A. Standard-of-care management was defined according to guidelines [23,24,25]. In practice, we considered patients as receiving standard of care when they recieved surgery for stages I and II, chemotherapy and surgery or radiotherapy for stage III, and chemotherapy or best supportive care in stage IV. LC–ILD and LC–noILD groups had comparable percentages of patients not managed according to standard-of-care recommendations (20.4% vs. 23.4%, respectively).

LC–ILD cases, compared to LC–noILD controls, respectively, experienced comparable rates of global treatment adverse events (48.9% vs. 35.2%; *p* = 0.09); had more frequent non-pulmonary infectious complications (12.2% vs. 2.8%; *p* = 0.01); pulmonary complications (25% vs. 23.7%), pulmonary infections (14.3% vs. 12.3%) and hematological adverse event rates (14.3% vs. 17.2%) did not differ. Pulmonary complications included three exacerbations (two after radiotherapy) and four infections. Pulmonary complication led to death in four patients with LC-ILD and none in LC-non ILD. Detailed stage, intervention and complication information for patients treated surgically (*n* = 13) or with radiotherapy (*n* = 8) are reported in Appendix A, respectively).

PFS did not differ between LC–ILD and LC–noILD (Figure 2A, *p* = 0.07). Median OS was significantly shorter for LC–ILD than LC–noILD patients (Figure 2B, *p* = 0.04). One- and two-year survival rates for LC–ILD and LC–noILD patients were 44.7% vs. 63.1% and 19.2 vs. 37.9% (*p* = 0.03 and *p* = 0.01), respectively.

Univariate and multivariate analyses to identify prognostic factors are reported in Table 3. Univariate analysis did not include WHO PS because of missing data (>20% unknown). Our Cox model included age, smoking history, body mass index, comorbidities (cardiovascular disease, chronic obstructive pulmonary disease and diabetes mellitus), ILD presence and standard-of-care LC management. Only ILD presence and standard-of-care LC management were independent factors retained as being significantly associated with risk of death.

Subgroup analyses of non-small–cell lung cancers (NSCLCs) (41 LC–ILD cases and 121 LC–noILD controls) found similar results, with a shorter median OS for LC–ILD than LC–noILD (Figure 2C). Multivariate analyses (HR [95% CI]) retained: age (1.02 [1.00–1.041]; *p* = 0.03), LC stages I–II vs. III–IV (0.14 [0.08–0.27]; *p* < 0.0001), ILD presence (2.02 [1.32–3.09]; *p* = 0.001) and standard-of-care LC management (0.43 [0.28–0.69]; *p* = 0.0004) as independent factors associated with risk of death.

## 4. Discussion

The ILD rate was 5.4% in this French LC-cohort study that included all LC stages. Compared to patients with LC–noILD, LC–ILD patients were significantly more frequently men, but the two groups did not differ in terms of age, smoking habits or tumor stages. LC–ILD histology showed a trend towards fewer adenocarcinomas. A majority of patients harbored very moderate functional impairment and radiological modifications, so we considered them to have mild ILD. Despite most patients having mild ILD with a mean VC over 80% and radiological extent around 5%, the LC–ILD prognosis was worse than that of LC–noILD. ILD presence and standard-of-care LC management were the only independent factors predictive of death. For NSCLC–ILD patients, age and stage were also prognostic factors.

Based on a few series, most of them from Japan and surgical series, ILD frequency in LC populations ranged between 2.4% and 10.9% [6]. Only one of three studies on European patients considered all LC stages and reported ILD frequency: 5.3% IPF patients among 637 LC patients from a single Irish institution; however, the clinical and biological investigations used to diagnose IPF were not given [17,18,26]. Our analysis on patients from two centers yielded a similar percentage (5.4%), thereby supporting this ILD rate in LC patients. Male sex is known to be a risk factor for LC in IPF patients and the male predominance found herein for LC–ILD compared to LC–noILD is also well-established [13]. This agrees with the predominance of men in IPF populations and among smokers undergoing low-dose CT screening for LC that detected ILD (9.2% vs. 4.3%) [21,27]. The influence of sex hormones on interstitial fibrosis development might explain this predominance [28].

Although LC-histology distribution varied in different published studies, LC–ILD patients had lower percentages of adenocarcinomas and higher frequencies of squamous-cell carcinomas than those with LC–noILD [7]. In our cohort, adenocarcinomas were less frequent and undifferentiated carcinomas more frequent, while the squamous-cell carcinoma rates were similar. The difficulty of obtaining adequate samples because of the pulmonary fibrosis and the paucity of tumor material probably prevented precise histological subtype characterization of some LCs, and thus the high undifferentiated-carcinoma frequency. As this couldn’t be explained by the presence of ILD, we supposed that clinicians, worried about complications such as pneumothorax or acute exacerbation, avoided some types of procedures, such as CT-guided biopsy. ILD’s presence could force clinicians to accept incomplete results because of the unfavorable benefit:risk ratio. Some series also reported impossible histological diagnoses for up to 20% of their patients [29].

LC–ILD treatment and prognosis have been studied mainly for specific LC stages and Asian patients. Regardless of LC stage and the treatment undertaken, excess mortality has been attributed to ILD-related complications, especially acute exacerbation, and cancer progression. Therefore, treating LC–ILD raises the issue of which standard of care should be applied to these patients with diminished respiratory reserves and increased risk of acute exacerbation. According to several reports on some LC–ILD series, fewer patients benefited from appropriate surgery or chemotherapy compared to LC–noILD patients [12,15,18,30,31]. The percentage of our cohort patients not receiving standard-of-care management did not differ between LC–ILD cases and LC–noILD controls. Pertinently, multivariate analysis retained standard-of-care management as an independent factor predictive of increased risk of mortality.

Several case–control studies on Asian IPF patients, mostly unmatched, have been published. Authors of studies on surgically treated LC patients reported that IPF or ILD was associated with shorter survival and that the presence of either pattern was an independent prognostic factor [32,33,34,35]. For chemotherapy-treated patients, IPF or idiopathic ILD was an unfavorable factor for survival [15,36]. Nishino et al. observed that 17 of 120 stage-IV NSCLC patients followed in a single institution in Boston, MA, had interstitial lung abnormalities that were associated with shorter survival [37]. We wondered whether those findings could be extended to a larger Caucasian population with all types of ILD and subclinical ILD. To investigate the prognosis and identify prognostic factors, we conducted a matched case–control study based on the results of the analysis of the whole LC population. In our cohort, 39% of patients had IPF, 20.4% had a UIP radiological pattern and most patients had mild ILD. Despite these characteristics, our case–control analysis confirmed that ILD was an independent factor associated with greater risk of death. Median OS was significantly shorter for LC–ILD than LC–noILD for patients and the case–control subgroup with NSCLC. We don’t have available mechanistic data that could explain this poorer prognosis. However, we suggest that the more frequent deaths due to pulmonary complications after the first line treatment, as well as the lower treatment intensity, could be an explanation. Indeed, the proportion of patients receiving more the two lines of treatment seemed to be lower in LC-ILD than in LC-noILD patients (14% vs. 26%, data not shown).

Curative LC treatment, based on surgery and radiotherapy, should be carefully discussed for patients with associated ILD. For surgical LC–ILD patients, the risks of acute exacerbation and cancer progression primarily depend on preoperative PFT results and the resection volume. According to Voltolini et al., operable patients with FVC > 90% did not develop postoperative acute respiratory insufficiency [26]. Yano et al. reported that limited resections for small-sized tumors were associated with a 2.78-times higher recurrence risk [38]. Only 12/49 of our cohort patients were treated surgically (see Appendix A). Comparing LC–ILD to LC–noILD patients, respectively, infectious complications (25% vs. 17%) and postoperative deaths (16.7% vs. 4.9%) were more frequent but did not reach statistical significance because of too few patients. When severe ILD is present, radiotherapy for inoperable LC is contraindicated because it causes radiation pneumonitis in up to 43% of patients [39]. Subclinical ILD is also considered a radiation-pneumonitis risk factor with some attributable fatalities, hence, the indication of radiotherapy, even stereotactic, should be decided by carefully considering the risks and benefits for ILD patients [40]. In our cohort, eight LC–ILD patients underwent radiotherapy: two (25%) developed non-severe radiation pneumonitis, statistically comparable to LC–noILD (12%) patients (see Appendix A).

Our study has several limitations. First is its retrospective design, meaning that some patients could not be included because of missing data, but only a few were not included because of poor-quality CT scans and most patients not included did not meet selection criteria. Because we investigated an LC population, CT scans were not acquired to study the parenchyma, but the majority had good-quality CT scans with thin slices. The design also prevented us from determining response rates and identifying causes of death. Finally, this study was done in 2012 and 2013, and did not include patients treated with immune checkpoints inhibitors. Those agents are now essential LC treatments and their use in the context of ILD is a very important issue [41].

Our study has several strengths. First, it is one of the largest to be reported to date on LC–ILD in a Caucasian population. Second, the centralized CT-scan review by two ILD experts assured accurate assessment of ILD patterns. Third, the matching of cases and controls on factors associated with poorer survival is a major factor helping to properly analyze the prognostic impact of ILD, independently of well-known poor-prognosis factors. Moreover, the main outcome measure in this study was death, which is a major event easily collected even with a retrospective design.

In conclusion, based on a large cohort of LC patients from two French centers, 5.4% had concomitant ILD. Despite mild ILD, the prognosis was worse, carrying a 1.8 higher mortality risk than for LC–noILD patients. ILD should be carefully sought in LC patients and taken into consideration when selecting among treatment options. Further studies are needed to determine the best therapeutic strategy.

## Figures and Tables

**Figure 1 jcm-09-00700-f001:**
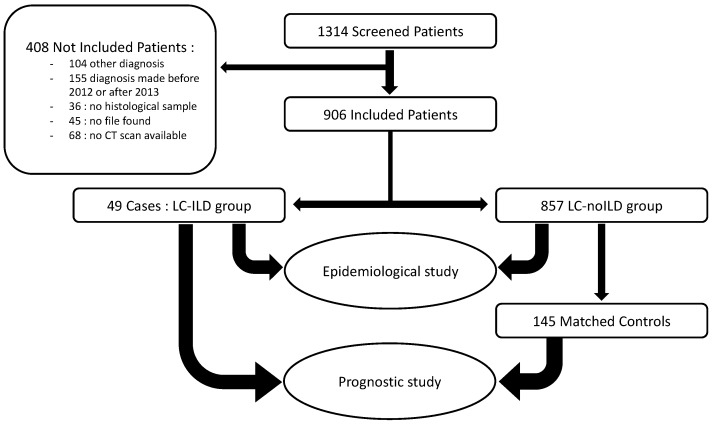
Flow chart: patient selection from screening to epidemiological and prognostic study.

**Figure 2 jcm-09-00700-f002:**
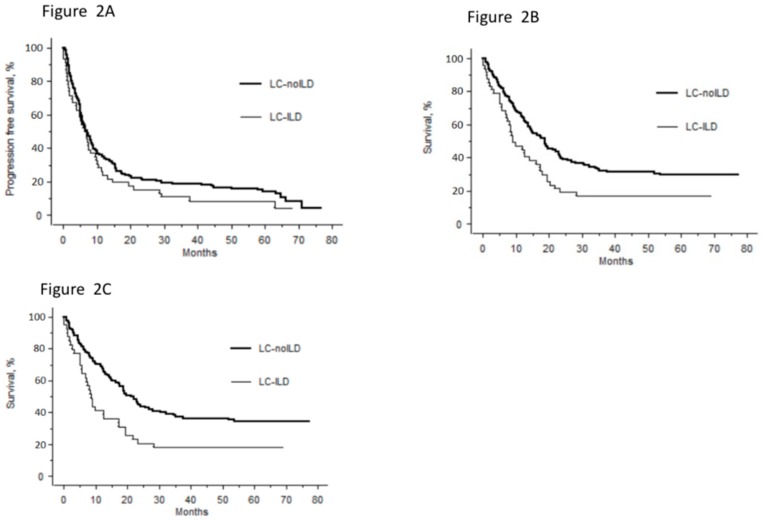
Survival rates of lung cancer (LC) patients as a function of interstitial lung disease (ILD) status. (**A**) Median (range) progression-free survival was 6.9 [0.1–68.2] for LC-ILD group vs. 7.1 [0.7–76.7] months for LC-noILD patients (*p* = 0.07). (**B**) Overall survival (OS) rates for LC-ILD and LC-noILD patients, respectively 9 vs 17.5 months (HR: 1.8 [95% CI: 1.21–2.67]; *p* = 0.04). (**C**) According to our subgroup analysis of non-small-cell lung cancers among case-control study, OS rates for LC-ILD cases vs. LC-noILD controls, respectively, were 8.2 vs 18.8 months (HR: 2.02 [95% CI: 1.32–3.09]; *p* = 0.001).

**Table 1 jcm-09-00700-t001:** Comparison of LC–ILD and LC–noILD Cohort-Patient Characteristics.

Characteristic	LC–ILD (*n* = 49)	LC–noILD (*n* = 857)	*p* Value
Males, *n* (%)	41 (83.7)	567 (66.2)	0.017
Mean age at diagnosis, years, ± SD	66.4 ± 8.8	64.7 ± 11.3	1
Smoking history			
Non-smoker/ever-smoker, %	8.2/91.8	12.7/87.3	0.47
Current smoker/ex-smoker, %	60/40	60.3/39.7	1
Mean pack-years, ± SD	44.4 ± 22.0	45.4 ± 25.9	1
Performans status: 0–1/2–4/U, %	47/25/29	59/20/22	0.19
Asbestos: NE/U/ARW/E, %	59.2/18.4/6.1/16.3	59.4/17.0/10.7/12.8	0.70
Lung-cancer histology, *n* ^a^	51	866	0.08
Adenocarcinoma, %	47.1	58.7	
Squamous carcinoma, %	19.6	19.6	
Undifferentiated carcinoma, %	13.7	4.5	
Small-cell carcinoma, %	15.7	12.0	
Others ^b^, %	4.0	5.2	
Lung cancer stage, %			
I/II/III/IV NSCLCs	20.9/11.6/20.9/46.5	14.4/8.1/22.0/55.4	0.35
LS/ES SCLCs	50/50	30.8/69.2	0.50
Synchronous LC, *n* (%)	2 (4.1)	9 (1.1)	0.1
Mutation analysis, *n* subjects ^c^	23	438	0.56
Unknown status, *n* (%)	6 (26.1)	63 (14.4)	
Wild-type, *n* (%)	10 (43.5)	172 (39.3)	
Mutation+, *n* (%)	7 (30.4)	199 (45.4)	
EGFR/KRAS/ALK, *n* (%)	1 (4.3)/4(17.4)/1 (4.3)	59 (13.5)/101(23.1)/20 (4.6)	
Rare mutations *, *n* (%)	1 (4.3)	23 (5.3)	

Performans status was categorized in: 0–1, 2–4 or U for unknown; NE/U/ARW/E, not exposed/unknown/at-risk worker/exposed; NSCLC, non-small–cell lung cancer; SCLC, small-cell lung cancer; LS, limited-stage; ES, extensive-stage; EGFR, epidermal growth-factor-receptor; KRAS, Kirsten rat-sarcoma viral oncogene; ALK, anaplastic lymphoma kinase; ^a^ Eleven patients had synchronous LC (2 with LC–ILD and 9 with LC–noILD). ^b^ Not done in 14 LC–ILD and 36 LC–noILD patients. ^c^ Only for advanced-stage adenocarcinomas. * Rare mutations: LC–ILD group: one BRAF (v-RAF murine sarcoma viral oncogene homolog B) mutation; LC-noILD group: one BRAF, three cMET, one EGFR (exon 20), one EGFR L858R & T790M, one EGFR & KRAS, seven HER2 (human epidermal growth factor receptor 2), one KRAS & PI3K, two EGFR mutations in a later analysis, two PI3K, one RET and one ROS1.

**Table 2 jcm-09-00700-t002:** Case–Control Epidemiological and Oncological Characteristics.

Characteristic	LC–ILD Cases (*n* = 49)	LC–noILD Controls (*n* = 145)
Males, *n* (%)	41 (83.7)	121 (83.4)
Mean age at diagnosis, years, ± SD	66.4 ± 8.8	66.4 ± 11.5
Smoking		
Ever-smoker, %	92	89
Mean pack-years, ± SD	44 ± 22	49 ± 27
Mean body mass index, ± SD	23.8 ± 4.3	24.5 ± 4.9
Mean performance status, ± SD	1 ± 0.9	1 ± 1
Comorbidities		
Chronic obstructive lung disease	10 (20.4%)	21 (14.5%)
Diabetes	8 (16.3%)	17 (11.7%)
Cardiovascular	26 (53.1%)	63 (43.4%)
Lung-cancer histology, *n*	49	145
Adenocarcinoma, (%)	23 (46.9%)	69 (47.6%)
Squamous carcinoma, (%)	10 (20.4%)	30 (20.7%)
Undifferentiated carcinoma, (%)	6 (12.2%)	17 (11.7%)
Small-cell carcinoma, (%)	8 (16.3%)	24 (16.6%)
Others, (%)	2 (4.1%)	5 (3.4%)
Lung cancer stage, %		
I/II/III/IV NSCLCs	17/10/22/51	16/12/21/1
LS/ES SCLCs	50/50	50/50

NSCLC, non-small–cell lung cancer; SCLC, small-cell lung cancer; LS, limited-stage; ES, extensive-stage.

**Table 3 jcm-09-00700-t003:** Univariate and Multivariate Analyses of Risk Factors Associated with Death.

	Univariate		Multivariate (*n* = 177)		Multivariate Descending Stepwise (*n* = 193)	
Factor	*n*	*p*	HR 95% CI	*p*	HR 95% CI	*p*
Age	194	0.01	1.01 (0.99–1.03)	0.23		
Sex						
Female	32	0.31				
Male	162					
Body mass index	178	0.097	0.96 (0.92–1.00)	0.08		
Smoker						
No	20	0.05	Reference	0.01		
Yes	174		2.67 (1.22–5.81)			
Chronic obstructive lung disease						
No	163	0.4				
Yes	31					
Diabetes mellitus						
No	169	0.09	Reference			
Yes	25		0.57 (0.32–1.02)	0.06		
Cardiovascular comorbidities						
No	105	0.56				
Yes	89					
Interstitial lung disease						
No	145	0.059	Reference			
Yes	49		1.80 (1.21–2.67)	0.004	1.81 (1.24–2.64)	0.002
Standard-of-care management						
No	44	0.002	Reference			
Yes	149		0.61 (0.39–0.97)	0.04	0.5 (0.34–0.73)	<0.001

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
