# Peer review of "Interstitial Lung Disease Associated with Lung Cancer: A Case–Control Study"

_jcm, 2020, doi:10.3390/jcm9030700_

Round 1

Reviewer 1 Report

  1. Are all patients with lung cancer getting HRCT, is it standard of care?
  2. Were any patients diagnosed with ILD based on pathology or they were all clinical diagnosis?
  3. Once patients were found to have ILD, were they worked up further to determine the cause of ILD?
  4. Once ILD was found, were they treated for ILD or just for cancer?
  5. What were the stages of cancer and locations and were they comparable?
  6. Were the treatment for respective stages comparable?
  7. Was data on performance status available?
  8. Any data on supplemental O2 use.
  9. What do authors attribute poor outcomes to?
  1. Are patients dying more from ILD
  2. They have aggressive lung cancers (noting that SOC management can be very aggressive or not very aggressive)
  3. They are getting less aggressive treatments
  4. Not treating the ILD’s?
  5. As with any study mechanistic data would be important

As the authors suggested that majority of these patients has mild ILD, 30% not even picked up on first go around by radiologist, that shouldn’t explain ILD contributing to worse outcomes

Reviewer 2 Report

This is a retrospective observational study looking at the characteristics of patients who have combined lung cancer and ILD comparing to those with lung cancer alone. The authors describe the incidence and outcomes of patients with LC-ILD compared t those with no ILD

This study had a good number of patients identified through the lung cancer meetings.

The abstract is clear and summarises the study results.

Keywords: only three included. could also add retrospective, outcomes,

Introduction:

Paragraph 1 last sentence requires a reference

Paragraph 2 1st sentence Typo with reference number 6 not in correct form

The authors frame the introduction in the context of their study regarding lung cancer and ILD studies. The aims of the study are outlined.

Methods:

Ethics described. Consent not required. Statistics explained

Results

Figure 1. please elaborate on heading for figure 1. Flow chart showing what? The flow chart shows an epidemiology and prognostic study and this hasn't been defined or explained in the methods section. Please explain what these are in the methods section.

Results 3.1 Typo at end of paragraph one

Table 1 Needs reformatting as it is too big

Table 2 Needs formatting as too big. Also appears in the document without any mention in the previous text. Are there any p values for table 2?

Table 3 Needs reformatting as it is too big

3.3. Case control study. This first paragraph should be in the methods as describes the methods used to chose the case controls

Page 6 . Any p values for survival difference? It is in the figure but not in the main text. Please include

Please define what standard of care lung cancer management is: Is this no specific treatment ie best supportive care? I see appendix B explains this but it may help in the man text to describe what treatments patients received

Discussion:

Paragraph 1 how do you define mild ILD?

Line 28: You say the ILD wasm mild but that then it was difficult to get histological confirmation. Why was this? The difficulty of 27 obtaining adequate samples because of the pulmonary fibrosis and the paucity of tumor material 28 probably prevented precise histological subtype characterization of some LCs

On the whole good discussion around comparative studies and discussed strengths and limitaions well.

Generally I feel all tables in appendix need formatting and tiding up

Author Response

Please see the attachment named "reviewer 2-2". 

Round 2

Reviewer 1 Report

No further comments